# Molecular Positivity of Porcine Circovirus Type 2 Associated with Production Practices on Farms in Jalisco, Mexico

**DOI:** 10.3390/v16101633

**Published:** 2024-10-19

**Authors:** Alberto Jorge Galindo-Barboza, José Francisco Rivera-Benítez, Jazmín De la Luz-Armendáriz, José Ivan Sánchez-Betancourt, Jesús Hernández, Suzel Guadalupe Sauceda-Cerecer, Jaime Enrique De Alba-Campos

**Affiliations:** 1Programa de Doctorado en Ciencias de la Producción y de la Salud Animal, Universidad Nacional Autónoma de México (UNAM), Mexico City 04510, Mexico; aljogaba@gmail.com; 2Laboratorio de Virología, Centro Nacional de Investigación Disciplinaria en Salud Animal e Inocuidad, Instituto Nacional de Investigaciones Forestales, Agrícolas y Pecuarias (INIFAP), Cuajimalpa, Mexico City 04010, Mexico; 3Facultad de Medicina Veterinaria y Zootecnia, Universidad Nacional Autónoma de México (UNAM), Mexico City 04510, Mexico; delaluzarmendarizj@fmvz.unam.mx (J.D.l.L.-A.); ivan.sanchez@posgrado.unam.mx (J.I.S.-B.); 4Laboratorio de Inmunología, Centro de Investigación en Alimentación y Desarrollo, A.C., Hermosillo 83304, Sonora, Mexico; jhdez@ciad.mx; 5Grupo Estatal de Vigilancia Epidemiológica, Comité Estatal para el Fomento y Protección Pecuaria del Estado de Jalisco, S.C., El Salto 45690, Jalisco, Mexico; suzelsauceda@gmail.com; 6Unión Regional de Porcicultores de Jalisco, El Salto 45680, Jalisco, Mexico; presidencia@urpj.org.mx

**Keywords:** porcine circovirus 2, PCV2, production practices, Jalisco, Mexico

## Abstract

The modernization of pig production has led to increasingly larger populations of pigs. This dynamic allows for accelerated production and ensures a steady pork supply but also facilitates the spread of infections. PCV2 is a ubiquitous virus and can cause PCV2-associated diseases, depending on production practices. This study aimed to evaluate the conditions of pig production in the state of Jalisco, Mexico, and correlate them with PCV2. A total of 4207 serum samples from 80 farms were analyzed. Epidemiological data were collected and used to investigate factors associated with PCV2 detection. A relative frequency of approximately 30% was detected, primarily in grower pigs maintained on multisite farms. Several production practices, particularly biosecurity measures, were associated with PCV2 on the analyzed farms.

## 1. Introduction

Porcine Circovirus Type 2 (PCV2), a member of the *Circoviridae* family, is the smallest swine virus, with a diameter ranging from 17 to 20 nanometers. It is a nonenveloped virus with a circular, single-stranded DNA (ssDNA) genome of approximately 1.7 kilobases in length. PCV2 is recognized as one of the most economically significant pathogens affecting swine worldwide. It is associated with a range of clinical manifestations that are collectively known as porcine circovirus-associated diseases (PCVADs), which include, but are not limited to, postweaning multisystemic wasting syndrome (PMWS), porcine dermatitis and nephropathy syndrome (PDNS), porcine respiratory disease complex (PRDC), and reproductive failure [1].

PCV2 is resistant to temperature and chemicals [2], allowing it to persist in the environment for extended periods. Studies have shown that even locations without pigs, such as the main entrance to the farm, can harbor viral loads ranging from 3.4 × 10^2^ to 3.8 × 10^5^ copies/swab of the virus in farms with systemic and subclinical disease, respectively, generating reservoirs and sources for the maintenance and reinfection of the virus [3].

Transmission occurs horizontally through direct contact between infected and susceptible pigs or indirectly via contaminated fomites [4]. The virus primarily targets immune cells, leading to immunosuppression and facilitating secondary infections [5]. Controlling PCV2 requires a multifaceted approach, including vaccination [6], biosecurity measures, and management practices aimed at reducing stress and maintaining herd health [7].

Globally, nine genotypes of PCV2 have been identified and classified (PCV2a- PCV2i). Historically, PCV2a was the predominant genotype; however, over time, PCV2b has been prevalent in many regions, including Mexico [8,9,10]. PCV2c was first detected and reported in Denmark in 2008, subsequently identified in Brazil [11,12], and later detected in China in 2016 [13]. PCV2d emerged around the year 2000 [14]; since then, it has been associated with severe clinical disease manifestations in some swine populations in Europe and the Americas, becoming the most prevalent genotype associated with clinical cases of PCV2 [9,15,16].

Several factors contribute to the genetic variability of PCV2. A significant factor is the high mutation rate of the virus, which leads to the accumulation of genetic changes over time [17]. Additionally, selective pressures, such as host immunity and vaccination programs, drive the evolution of PCV2 strains by favoring the survival of variants that evade immune recognition or vaccine-induced protection [18]. The extensive global distribution of PCV2 further facilitates genetic exchange between different viral populations, leading to the emergence of novel recombinant strains [16]. Environmental factors, including farming practices and biosecurity measures, also shape the genetic diversity of PCV2 by influencing the transmission dynamics and persistence of the virus within swine populations [19]. Understanding the interaction of these factors is essential for elucidating the mechanisms driving PCV2 evolution and developing effective disease control and prevention strategies.

In 2022, Mexico ranked as the eighth-largest producer of pigs globally, with a registered inventory of 20,894,799 heads [20]. The swine industry plays a crucial role in the national economy, with the state of Jalisco standing out as the leading producer, generating 397,849.4 tons of pork in the same year [20]. The region is characterized by advanced porcine infrastructure, high productivity, and the capacity to export high-quality pork products. However, it also includes family-based systems and backyard farms (1800 of these units were recorded in 2021), reflecting the diversity and complexity of the swine sector in relation to the establishment of disease control measures [21].

In Mexico, there are limited data on the incidence of PCV2 infection associated with various production practices. This study uses Jalisco as a model to correlate these conditions with PCV2 occurrence. Additionally, this study aims to generate data to support the development of strategies to improve the management and surveillance of PCV2, according to the specific characteristics of swine farming in regions with conditions similar to those of Jalisco.

## 2. Materials and Methods

### 2.1. Research Site

The study was conducted in Jalisco, Mexico. According to the *Union Regional de Porcicultores de Jalisco* (URPJ), Jalisco state is stratified into 4 regions that are differentiated by their pig population density [21].

### 2.2. Samples

In this study, 4207 serum samples were collected from 80 farms and analyzed at the Virology Laboratory of INIFAP (Palo Alto, Mexico City, Mexico).

The farms and samples were grouped into four clusters for analysis:Regions were categorized by their pig density per square kilometer and coded as A, B1, B2, or B3.Farm types were classified on the basis of the number of breeding sows: semi-intensive (21–500 sows) and intensive pig farming (≥500 sows).Farm types were also categorized according to the farm system: In farrow-to-finish commercial farms (FFs), all the stages of the pig production cycle, from farrowing to finishing, occur within a single site. In multisite pig farms (MSs), the different stages of production—reproduction (Site-1), transition (Site-2), and finalization (Site-3)—occur at separate locations.The production stage corresponds to specific age ranges: from birth to weaning (suckling phase), from weaning to 10 weeks (weaner phase), 11–14 weeks (growing phase), 15–18 weeks (grower phase), 19–22 weeks (finisher phase), and pregnant sows.

The number of farms distributed by region in the study was as follows: 34 in Region A, 10 in Region B1, 18 in Region B2, and 18 in Region B3. Up to 60 serum samples were collected from each farm, corresponding to the productive stages by age range (10 samples per stage), provided that the participating farm had inventory across all the production stages and considering whether it was an FF or an MS farm. Pigs without clinical signs and pigs with clinical signs that were compatible with PCV2 infection (growth retardation, respiratory distress, diarrhea, skin discoloration, jaundice, weight loss, and reproductive failure) were sampled. Among the 4207 samples, 3802 corresponded to pigs without clinical signs, and 405 corresponded to pigs with clinical signs. The study population, distributed by region and the age group of the pigs (or production stage) as described above, is presented in Table 1.

### 2.3. Real-Time PCR

The samples collected from each farm were pooled into groups of five per production stage, with two pools per stage. Thus, for the FFs, 12 pools were obtained, representing the 6 production stages. For farms lacking all the production stages (e.g., MS farms), only samples from the available stages were collected. In total, 844 pools were obtained from the 4207 samples, which were clearly identified at the farm and production stages. When the pig inventory per production stage was limited, the samples were pooled with <5 serum samples.

The DNA from each pool was extracted using the QIAwave DNA Blood & Tissue Kit (QIAGEN, Hilden, Germany, Cat. No. 69556), following the manufacturer’s recommendations. The DNA samples were stored at −76 °C for further analyses. A real-time PCR (qPCR) analysis to detect ORF2 was performed via the QuantiTect Probe PCR Kit (QIAGEN, Cat. No. 204345). A qPCR was performed in a final volume of 10 µL/reaction. Five microliters of 2x QuantiTect Probe PCR Master Mix, 1.25 µL of RNase-free water, 2 µL of DNA (≤500 ng/reaction), and 1.2 µM primers, and a probe in 1.75 µL were used (Table 2).

A qPCR amplification was conducted in a CFX96™ Real-Time System (Bio-Rad Laboratories, Inc., Hercules, CA, USA) thermocycler as follows: initial denaturation at 95 °C for 15 min and 40 cycles at 94 °C for 15 s and 60 °C for 1 min. The cycler results were analyzed with Bio-Rad CFX Manager 3.1 software (Bio-Rad Laboratories). The Ct was defined by applying a fluorescence drift correction and using a positive control to confirm that the threshold was the same in all the experiments. Samples with Ct values > 35 were considered negative.

### 2.4. Data Collected

A survey was conducted on the farms to collect data regarding pig production practices, including production management, feed and water provisions, disinfection and fomite control, sanitary management, facility and equipment standards, pest management strategies, and waste disposal methods. The above information was used to analyze the factors associated with PCV2 infection (Table 3). The statistical analysis is described in the section entitled “Statistical analysis”.

### 2.5. Statistical Analysis

The data collected from surveys conducted on 80 farms were used to construct a database. Each column in the database represents potential risk factors, whereas the variable of interest is the qPCR result indicating the presence or absence of PCV2. The rows corresponded to individual observations, specifically the analyzed pools.

To analyze the clusters, 2 × 2 contingency tables were generated, and the chi-square test was employed with a significance level of α = 0.05 to assess the independence of the variables. Yate’s correction was applied in cases with fewer than ten observations per group. For factors that demonstrated a statistically significant association (*p* < 0.05), odds ratios (ORs) and 95% confidence intervals (CIs) were calculated. Factors found to be independent of the presence of PCV2 in any of the analyzed strata were excluded from further analysis.

The groups showing risk factors were used as the reference group for calculating the OR. Identifying these groups in the presence of risk allowed for quantitative comparisons between the study groups, enhancing the interpretation of the magnitude and direction of the observed associations. This analytical approach ensures that the calculated OR accurately reflects the relative probability of the event of interest, thereby providing a robust basis for interpreting the results. The statistical analyses were performed via RStudio (version 2023.06.0) via R (version 4.3.1), which uses the “epibasix” package for elementary epidemiological and biostatistical functions.

## 3. Results

The pig density in each region was 9.46, 210.36, 261.65, and 135.4 pigs/km^2^ in regions A, B1, B2, and B3, respectively. The regions were delimited geographically and by animal density, as shown in Figure 1.

The relative frequency (RF) of PCV2 per region detected by the qPCR was 39.9%, 37.7%, 24%, and 18.39% in areas B2, B1, A, and B3, respectively, from the highest to lowest frequency. Table 4 displays the relative frequency by production stage and farm type, based on the number of breeding sows raised. For the semi-intensive pig farms studied, an RF of 20.6% was obtained, with the highest RF recorded in Region B1, at 36.1%. Conversely, an RF of 44.7% was obtained for the intensive pig farms, with Region B2 showing the highest frequency, at 60.39%.

The study included five Site-1 farms and seven Site-3 farms that were classified as MSs. FFs comprised 80% of the farms included in the study. The RF of PCV2 occurrence was 26.3% on the FF farms and 43.8% on the MS farms. Among the MS farms, the Site-3 farms had a presentation rate of 56.5%, whereas the Site-1 farms had a presentation rate of 22.2%. Since the majority of the pig farms in the study were classified as FF farms, the RFs for this farm type were as follows: Region A—24.5%; Region B1—36.3%; Region B2—35.0%; and Region B3—16.6%. Additionally, for the FF farms, the relationship between the regional factor and the presence of PCV2 is statistically significant (χ^2^ = 18.9035; *p* = 0.000286).

Table 5 presents the risk factors associated with PCV2 identified through the chi-square test. It also includes the odds ratio, which indicates the likelihood of PCV2 positivity in the groups that were exposed to the risk factor compared with those not exposed.

## 4. Discussion

In this study, we investigated the associations between specific biosecurity practices, risk factors, and the presence of PCV2, detected through viral DNA via qPCR, hereafter referred to as viremia. Our results revealed significant associations between several risk factors and viremia, supported by X^2^ values and odds ratios (ORs). The key findings are discussed below. Although technical reports on biosecurity measures have been available for some time [23,24], few studies have focused on their effectiveness in reducing the risk of disease introduction [25,26]. Notably, 75% of the farms studied (60 out of 80) had at least one sample positive by qPCR, classifying them as PCV2 positive. This finding indicates a high prevalence of the virus in the Jalisco region. Among the farms studied, 88.7% (71/80) reported using vaccination during the nursing–weaning stages as their primary disease control measure. In contrast, in Jalisco, only 12.4% of commercial and backyard farms vaccinate (n = 2213), whereas 33% of semi-intensive farms and 50% of intensive farms adopt vaccination practices [21]. Therefore, adopting this control strategy among the studied farms can be considered high.

Commercial pig farms are reported throughout the territory of Jalisco; however, their distribution is not homogeneous; there are regions where the concentration of farms is relatively high, consequently increasing the population density, as is the case in Region B2 [21]. When we analyzed positive pools by region in Jalisco, we found that Regions B2 and B1 had the highest relative frequencies (RFs), at 39.9% and 37.7%, respectively. This aligns with the fact that these regions also had the highest animal densities (261.65 and 210.36 pigs/km^2^). Additionally, Region B2 had the largest number of intensive farms and the highest breeding stock inventory in Jalisco [21]. In contrast, Region A, which had the lowest animal density (9.46 pigs/km^2^), presented a relative frequency (RF) of 24%. Although this is not the lowest frequency observed in this study, Region A had the highest number of backyard farms and the lowest number of commercial farms in Jalisco, where limited vaccine use has been reported. For example, in the absence of data on the percentage of vaccination against PCV2, other diseases, such as PRRS, blue eye disease, and influenza, show that 94.2%, 99.1%, and 96.4% of farms in this region, respectively, do not vaccinate [21]. Given these characteristics, the frequency of PCV2 in Region A could be associated with the presence of disease reservoirs, which may explain the observed disease patterns in Jalisco.

In Jalisco, there are different types of farms, which are classified according to the number of sows. Intensive farms are mostly concentrated in Regions B1 and B2, whereas semi-intensive farms are present throughout the state. These results revealed that intensive farms had the highest virus detection frequency in Region B2 (60.39%), while semi-intensive farms had a rate of 36.1% in Region B1. These findings suggest that farm type and location, which are associated with density, may influence the presence of PCV2.

The frequency of PCV2 occurrence also appears to be partially related to the type of production system used by the farms. For the FFs, the RF was 26.3%, whereas for the MSs, the RF was 43.8%. This trend can be explained by analyzing the production stages maintained in each type of farm. MS farms are particularly relevant to this study, as an RF of 22.2% was observed at Site-1 and 56.5% at Site-3. Although the number of MS farms observed was minimal, the results indicate that the production stage with the highest RF (39.5%) was the 15–18-week age group, which was raised at Site-3.

According to the distribution of samples by production stage, the pregnant sows had the lowest RF, 14.3%. Some studies have reported that viremic sows are not present on positive farms [27], in contrast with other findings that reported an RF of 47.2% in pregnant sows from farms in Mexico and the United States of America [28]. Piglets often become infected at early ages, with clinical signs of the disease appearing after weaning [29], which may be related to the high RF observed in the growing, grower, and finisher pigs (37.3%, 39.5%, and 31.7%, respectively), particularly when risk factors and virus reservoirs are present in the facilities housing these animals.

Now, we address the risk factors identified by category. In those related to production management, the farms in Region B3, FF farms, and semi-intensive systems that introduce replacement gilts without implementing sanitary protocols had 5.7, 1.5, and 1.9 times higher odds of presenting viremia, respectively, than those with strict biosecurity protocols. This finding is consistent with the study by Afolabi et al. (2023), which demonstrated that farms lacking quarantine and replacement generation protocols are more likely (OR = 4.4) to test positive for PCV2 than those that follow these protocols. Particularly notable are the farms in Region B3, where 80.3% of the analyzed samples originated from farms without strict biosecurity measures for the entry of replacement gilts and boars [30].

In the “disinfection and fomites” category, the farms in Region B2 and intensive production systems presented a significantly greater likelihood of viremia (OR = 21.5 and 15.0, respectively) due to the inadequate control of access than did those implementing such measures. This suggests that the high density of pigs and the intensity of production on farms with large pig inventories can facilitate viral transmission by permitting entry without proper sanitary control, thereby promoting the presence of the virus. Similarly, the failure to disinfect vehicles entering these farms was associated with 19.3- and 15-fold greater odds of testing positive for PCV2 than in farms that followed disinfection protocols. Another notable finding is that the semi-intensive farms, where workers move between different production areas, and the farms in Region A, where veterinarians and technicians visit multiple farms, faced 5.5- and 1.8-fold greater risks, respectively, than farms that mitigate these risks. Importantly, the farms in Region A and the semi-intensive farms often lack dedicated technical personnel, resulting in the same staff managing various production areas, which exacerbates the spread of disease.

Several reports have described the importance of biosecurity protocols to prevent the introduction or recirculation of the virus [31,32,33,34]. Previous studies have reported high viral loads in parking areas (4.2 × 10^3^ PCV2 copies/swab) and even in the hands of personnel (4.6 × 10^3^ PCV2 copies/swab) and other individuals entering farms [3]. Moreover, the farms that restrict visitor access had a lower likelihood of infection (OR = 0.122), whereas the farms without such restrictions were 8.19 times more likely to have positive pigs [30]. Although this study did not find a significant association between the risk factors related to boots and clothing and viremia, previous research indicates that these factors can be sources of infection, with viral loads as high as 2.9 × 10^5^ PCV2 copies detected in farms with subclinical infections [3]. This finding highlights the importance of controlling access and disinfecting vehicles to mitigate the infection risk.

For the “health management” category, in the farms located in Region B2 and the intensive farms, the absence of a deworming program significantly increased the likelihood of viremia (OR = 7.8 and 3.4, respectively). A similar study in Africa indicated that farms with a history of coccidiosis are 14.3 times more likely to be infected with PCV2 than those without this parasite [30].

The lack of specific vaccines, such as those formulated against *Erysipelothrix rhusiopathiae*, in the farms from Region B2, intensive farms, or closed-loop systems, is associated with an increased presence of PCV2 (OR = 8.5, 2.9, and 1.7, respectively). Furthermore, the intensive farms that do not administer an influenza vaccine (OR = 2.6; 1.5–4.6) also showed a greater association with the presence of PCV2. Previous studies have demonstrated that the influenza virus can coinfect pigs and is linked to porcine multisystemic wasting syndrome [35]. Additionally, the fattening stage of pigs, particularly those between 19 and 22 weeks of age, is compromised when they are not vaccinated against erysipelas, with the probability of viremia being 3.8 times greater than that of vaccinated pigs. Moreover, the risk factors associated with the presence of PCV2 were identified in two production stages. During the fattening stage, in pigs aged between 11 and 14 weeks, the absence of vaccination schedules against influenza, *Glaserella parasuis*, PRRS, and erysipelas increased the likelihood of viremia by 2.1- to 3.5-fold. However, when pigs in this age group were not vaccinated against PCV2, the risk increased 5.7-fold.

The farms in Region B3 presented the highest risk of viremia when the pigs were not vaccinated against PCV2, with the likelihood being 6.0 times greater than that associated with vaccinated farms. According to the Epidemiological Surveillance Group in Jalisco, this region is a transitional zone with high pig mobility due to marketing and distribution to other areas, representing a potential infection risk. Moreover, within this region, 60.1% of commercial and backyard farms (N = 751) do not vaccinate against PCV2 [21], further exacerbating the risk. Notably, the farms in Region B1 were not exposed to this risk factor, as all the farms reported the use of PCV2 vaccination for disease prevention. Although positive samples were detected, a numerical association between viremia and risk could not be established. However, it is well known that vaccination against PCV2 in weaned piglets may be associated with the presence of the virus in serum [36]. These findings align with numerous studies that emphasize the critical importance of vaccinating pigs around the weaning stage to mitigate challenges in later production stages [37,38,39]. Additionally, the absence of protection against other pathogens and the presence of coinfections increase the risk of PCV2 viremia [40,41]. Furthermore, inadequate biosecurity and a lack of protective measures exacerbate the occurrence of clinical signs and multietiological conditions.

On Multisite farms that do not have services outside them (such as gas loading points, electrical stations, and product reception areas, which help restrict access to the farm) exhibited up to 13-times-higher odds of PCV2 viremia than those with such external services did. Other clusters showing similar risks included the semi-intensive farms and those in Region B3, as previously discussed, likely due to the unique production practices in these units. In Region B1, which is located in the northeastern part of the state, the presence of feral pigs is associated with a 9.1-fold increased likelihood of PCV2 viremia compared with regions where feral pigs have not been observed. Although additional factors may contribute to this condition, the presence of PCV2 and other pathogens has been reported in feral pigs in Mexico [42], and when these animals approach farms, they could pose a transmission risk.

The results underscore the critical importance of proper waste management. In this category, the failure to adequately handle carcasses on the Region B2 farms, intensive farms, and FF farms was associated with a significantly greater likelihood of viremia (OR = 21.5, 15, and 2.2, respectively). Some studies suggest that improper carcass disposal leads to elevated microbial levels, creating reservoirs where vectors and fomites serve as the primary mechanisms for virus reintroduction [43]. In the Region B1 farms, improper wastewater treatment represents a significant risk, with a greater likelihood of being PCV2 positive (OR = 29) than farms that manage waste effectively. This region is characterized as one of the driest in Jalisco, where untreated wastewater is frequently used for crop irrigation, thereby exacerbating the spread of pathogens.

Finally, the lack of association between certain factors and the presence of PCV2 viral DNA in serum can be attributed to the retrospective cross-sectional design of this study, which may have complicated the numerical analysis in the absence of well-established control groups. Nevertheless, relevant inferences were made that align with the expected outcomes. Prospective case‒control studies are needed to provide concrete evidence and quantify the effects of the most significant risk factors. Moreover, these findings emphasize the need to consider these factors in swine disease control and prevention strategies.

## 5. Conclusions

The present study identified the primary production practices associated with PCV2 in farms located in Jalisco, Mexico. The prevalence in different production systems allows the identification of critical points, such as those related to sanitary management, access control, and waste management. Strategies to improve swine production will favor the adequate control of PCV2 infection and increase swine production.

## Figures and Tables

**Figure 1 viruses-16-01633-f001:**
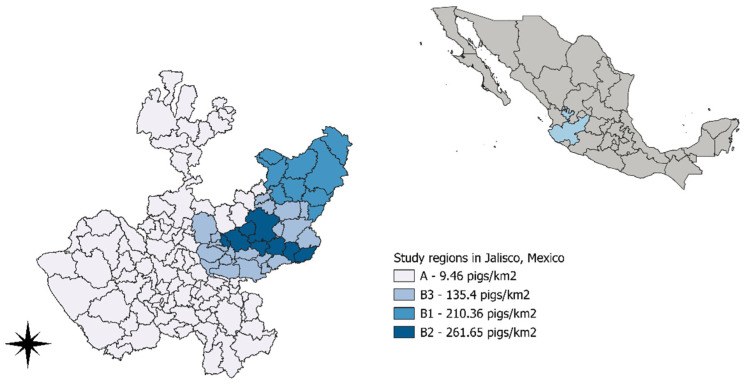
Representative location of Jalisco, Mexico, the sampled areas, and the pig population density (pigs/km^2^).

**Table 1 viruses-16-01633-t001:** Distribution of the study pig population by region and age range.

Age Range of the Pigs	Region			
	A (n = 1924)	B1 (n = 571)	B2 (n = 832)	B3 (n = 880)
Birth to weaning	276	76	128	122
Weaning to 10 weeks	344	120	126	150
11 to 14 weeks	383	121	139	168
15 to 18 weeks	344	82	161	169
19 to 22 weeks	275	86	136	127
Pregnant sows	302	86	142	144

**Table 2 viruses-16-01633-t002:** Primers and probes used for PCV2 real-time PCR.

Primers and Probe	Sequences (5′ → 3′)
PCV2F [22]	CCAGGAGGGCGTTGTGACT
PCV2R	CGCTACCGTTGGAGAAGGAA
PCV2-Probe	FAM/AATGGCATCTTCAACACCCGCCTCT/BHQ1

**Table 3 viruses-16-01633-t003:** Possible general risk factors identified from the survey application to pig farms: classification and coding for analysis in the present study.

Category	Risk Factor	Codification
Production management	Failure to implement the “All in, All out” system	all
	Purchase and entry of gilts and boars without complying with animal health protocols.	gilts
	Not on-farm replacement gilt production	generate
Feeding and water provisions	Feeding with scraps food (swill feeding).	swill_feeding
Disinfection practices and fomite control	Lack of farm access control	access
	No change of clothing upon farm entry	clothing
	No change of boots or clothing between areas	boots
	Failure to disinfect vehicles upon farm entry	vehicles
	Failure to disinfect equipment and work material	material
	Failure to wash pens	pens
	Technical staff visits other farms	technical
	Workers visit other farms	workers
	Workers move between production areas	move
	Absence of sanitizing mats	mats
	Failure to disinfect pens	disinfect
Sanitary management	Failure to deworm	deworm
	Failure to vaccinate against influenza	flu
	Failure to vaccinate against blue eye disease	blueye
	Failure to vaccinate against *Bordetella* + *Pasteurella*	bp
	Failure to vaccinate against *E. coli*	coli
	Failure to vaccinate against *Glaserella parasuis*	parasuis
	Failure to vaccinate against PRRSV	prrs
	Failure to vaccinate against PCV2	circo
	Failure to vaccinate against Parvovirus, Leptospira, and Erysipelas on sows.	ple
	Failure to vaccinate against *Mycoplasma*	myco
	Failure to vaccinate against *Actinobacillus pleuropneumoniae*	app
	Failure to vaccinate against Erysipelas only	ery
Facility and equipment standards	Absence of perimeter fence	fence
	Loading docks of pigs are inside the farm	docks
	Services not located outside the farm (water intake, energy, gas, reception of farm inputs).	services
	Absence of a specific dining room and bathrooms for collaborators within the farm	rooms
Pest management strategies	Failure to control rodents	rodents
	Failure to control insects	insects
	Collaborators also raise and care for pigs off-farm	off-farm
	Presence of wild pigs near farms	wild
Waste disposal methods	Failure to treat solid waste	waste
	Improper disposal of carcasses	carcass
	Failure to treat wastewater	wastewater

**Table 4 viruses-16-01633-t004:** Analysis of the frequency of PCV2-positive pools and their distribution across different productive stages in semi-intensive, intensive, farrow-to-finish farm, and multisite pig farming systems.

Age Range of the Pigs	Pools Positive/Analyzed (RF ^1^)				
	All Pools	Semi-Intensive (21–500 Sows)	Intensive (≥500 Sows)	Farrow-to-Finish	Multisite
Birth to weaning	19/123 (15.4%)	9/87 (10.3%)	10/36 (27.8%)	17/114 (14.9%)	7/9 (77.8%)
Weaning to 10 weeks	37/149 (24.8%)	21/111 (18.9%)	16/38 (42.1%)	33/136 (24.3%)	4/13 (30.8%)
11 to 14 weeks	60/161 (37.3%)	35/119 (29.4%)	25/42 (59.5%)	52/149 (34.9%)	8/12 (66.7%)
15 to 18 weeks	60/152 (39.5%)	34/100 (34%)	26/52 (50%)	52/137 (38%)	8/15 (53.3%)
19 to 22 weeks	40/126 (31.7%)	17/82 (20.7%)	23/44 (52.3%)	32/112 (28.6%)	8/14 (57.1%)
Pregnant sows	19/133 (14.3%)	6/92 (6.5%)	13/41 (31.7%)	17/123 (13.8%)	2/10 (20.0%)

^1^ RF = Relative frequency.

**Table 5 viruses-16-01633-t005:** The risk factors identified for each analyzed cluster, along with the chi-square (χ^2^) values and odds ratios (ORs), demonstrate the association and likelihood of PCV2 presence in the groups exposed to the risk factor compared with those that were not exposed at farms in Jalisco, Mexico.

Coded Risk Factors	Cluster	χ^2^	N	*p*	OR	CI (95%)	
						Lower	Upper
gilts	Region B3	11.12	152	0.0008	5.7	2.025	16.29
	Semi-intensive	6.12	566	0.0133	1.9	1.172	3.332
	Farrow-to-finish farm	4.15	735	0.0416	1.5	1.034	2.328
access	Region B2	13.37	157	0.0002	21.5	2.704	171.788
	Intensive	9.35	237	0.0022	15.0	1.911	118.592
	Farrow-to-finish farm	6.52	756	0.0106	3.2	1.367	7.825
vehicles	Region B2	11.79	147	0.0005	19.3	2.419	154.066
	Intensive	9.35	237	0.0022	15.0	1.911	118.592
move	Semi-intensive	12.34	575	0.0004	5.5	2.003	15.631
technical	Region A	6.14	377	0.0132	1.8	1.164	3.053
deworm	Region B2	10.38	157	0.0012	7.8	2.109	29.092
	Region B3	7.41	165	0.0064	4.5	1.611	13.096
	Intensive	6.23	237	0.0125	3.4	1.347	8.502
	Multisite	7.00	68	0.008	4.6	1.597	13.806
flu	Intensive	11.72	237	0.0006	2.6	1.539	4.569
	Pigs aged 11 to 14 weeks	6.5	159	0.0107	2.6	1.286	5.151
blueye	Region B1	11.46	114	0.0007	4.2	1.884	9.572
coli	Region B3	6.00	165	0.0142	2.9	1.302	6.476
	Multisite	4.00	68	0.045	3.5	1.149	10.463
parasuis	Region A	6.23	388	0.0125	2.1	1.196	3.651
	Pigs aged 11 to 14 weeks	4.64	159	0.0312	2.4	1.144	5.246
prrs	Region B3	4.07	165	0.0435	2.6	1.117	6.252
	Multisite	4.00	68	0.045	3.5	1.149	10.463
	Pigs aged 11 to 14 weeks	3.84	159	0.0499	2.1	1.046	4.039
circo	Region B3	9.89	165	0.0016	6	1.989	18.103
	Pigs aged 11 to 14 weeks	3.79	159	0.0516	5.7	1.151	41.900
ple	Region B1	3.87	114	0.0492	3.1	1.113	8.877
	Region B3	9.44	152	0.0021	4.7	1.811	12.428
	Semi-intensive	6.12	574	0.0133	2.3	1.229	4.333
	Multisite	4.00	68	0.045	3.5	1.149	10.463
	Pigs aged 11 to 14 weeks	8.49	156	0.0035	4.4	1.661	12.376
	Pigs aged 19 to 22 weeks	3.85	122	0.0498	2.8	1.073	7.436
myco	Region B3	6.84	165	0.0089	3.9	1.492	10.434
ery	Region B2	9.27	157	0.0023	8.5	1.922	37.652
	Semi-intensive	5.67	587	0.0172	2.2	1.178	4.223
	Intensive	11.43	237	0.0007	2.9	1.589	5.362
	Farrow-to-finish farm	6.02	756	0.0141	1.7	1.133	2.673
	Pigs aged 11 to 14 weeks	6.21	159	0.0127	3.5	1.389	9.762
	Pigs aged 19 to 22 weeks	3.61	124	0.0500	3.8	1.124	16.709
services	Region B3	7.69	152	0.0055	3.5	1.496	8.176
	Semi-intensive	17.73	513	0.0000	2.7	1.708	4.318
	Farrow-to-finish farm	8.74	654	0.0031	1.7	1.219	2.521
	Multisite	14.75	64	0.0000	13.0	3.32	51.652
wild	Region B1	8.11	114	0.0044	9.1	1.869	44.616
	Semi-intensive	6.44	587	0.0111	2.2	1.23	3.947
	Farrow-to-finish farm	3.78	756	0.0519	1.8	1.032	3.058
waste	Intensive	5.50	237	0.0189	2.0	1.159	3.593
carcass	Region B2	13.37	157	0.0002	21.5	2.704	171.788
	Intensive	9.35	237	0.0022	15.0	1.911	118.592
	Farrow-to-finish farm	5.16	756	0.0230	2.3	1.165	4.467
wastewater	Region B1	18.55	114	0.0000	29	3.775	222.768
	Semi-intensive	13.20	587	0.0185	2.2	1.438	3.279

CI = confidence interval; *p* = *p* value; N = observations.

## Data Availability

Data from the farms studied are not available due to privacy restrictions.

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
