# Peer review of "Molecular Positivity of Porcine Circovirus Type 2 Associated with Production Practices on Farms in Jalisco, Mexico"

_viruses, 2024, doi:10.3390/v16101633_

Round 1

Reviewer 1 Report

Comments and Suggestions for Authors

In this manuscript, authors employed the Jalisco as a model to correlate these conditions with PCV2 occurrence, and the associations between biosecurity measures considered as risk factors and the occurrence of PCV2 was investigated. These findings are somewhat interesting, but I consider that the present paper shows these major concerns:

1. Lines 55-61: Currently, the present study demonstrated that PCV2 can be divided into 8 genotypes (PCV2a-PCV2h), and PCV2d is the current predominant genotype in the world.

2. Lines 101-102: To ensure the effectiveness of the research, it would be highly necessary testing this study with not only blood samples since these clinical samples can contain greater amount of PCR inhibitors, but also including the nasal fluid, saliva and feces.

3. Lines 103-104 and 108-109: There is a discrepancy in the description: 60 serum samples were collected from each of the 80 pig farms, a theoretical total of 4,800 samples, but the total number of samples collected is 3,802.

4. Lines 120-122: How were 844 serum pools selected without detailed explanation. If it is a mixed test of samples from the same pig farm, it should be shown as Table 1.

5. After analyzing and locating the sequences of primers and probes used in this study (Table 2), the detection work could not be carried out correctly, and the data were wrong.

6. Line 203: It needs to check the description is accurate: p=<0.05.

7. The data in Table 5 requires proper analysis in manuscript.

8. Please review all statistical reporting in your manuscript. It is important that the statistical analyses used are suitable for your study and that all statistical reporting is correct.

Comments on the Quality of English Language

The manuscript should be reviewed by a native English speaker who is familiar with scientific writing.

Author Response

Comments 1: Lines 55-61: Currently, the present study demonstrated that PCV2 can be divided into 8 genotypes (PCV2a-PCV2h), and PCV2d is the current predominant genotype in the world.

Response 1: Thank you for highlighting this. We agree with your comments. Consequently, we have revised the text in lines 55-62 to clarify that nine genotypes have been reported, of which PCV2a, PCV2b, and PCV2c have had clinical relevance. Additionally, we emphasize the current importance of the PCV2d genotype. The revised manuscript shows the changes in red.

Comments 2: Lines 101-102: To ensure the effectiveness of the research, it would be highly necessary testing this study with not only blood samples since these clinical samples can contain greater amount of PCR inhibitors, but also including the nasal fluid, saliva and feces.

Response 2: Thank you for pointing this out. We acknowledge the comment regarding the importance of considering additional sample types to strengthen the results. However, this study is cross-sectional and includes a substantial number of cooperating farms, which restricts access and movement within these facilities. We are currently designing further studies that will allow us to control for independent variables (factors) to validate the results obtained and generate supplementary information to that presented here.

Comments 3: Lines 103-104 and 108-109: There is a discrepancy in the description: 60 serum samples were collected from each of the 80 pig farms, a theoretical total of 4,800 samples, but the total number of samples collected is 3,802.

Response 3: Thank you for pointing this out. In theory, 4,800 samples were to be collected; however, some cooperating farms did not have inventory across all production stages because they were not full-cycle operations, which limited the availability of certain groups (e.g., in weaning-to-finishing farms, we were unable to collect samples from suckling piglets or pregnant sows). Therefore, the total number of samples collected was 4,207.

It should be noted that in farms with pigs displaying clinical signs suggestive of PCV2 infection, 405 pigs were sampled, while the remaining 3,802 were apparently healthy, making the total number of samples equal to 4,207. We have updated the text in lines 103-108 and 110-114 of the manuscript to enhance clarity. The changes made are highlighted in red in the revised manuscript.

Comments 4: Lines 120-122: How were 844 serum pools selected without detailed explanation. If it is a mixed test of samples from the same pig farm, it should be shown as Table 1.

Response 4: Thank you for pointing this out. The 844 pools were formed from the 4,207 individual samples collected, with each pool identified by farm and production stage. Two pools were generated per production stage, resulting in 12 pools per farm for those with a full production cycle. In some cases, pools were formed with only 3 or 4 individual samples (instead of five) due to limited inventory on the farm. To clarify this, we have revised the wording in lines 122-128 of the manuscript. The changes made are highlighted in red.

Comments 5: After analyzing and locating the sequences of primers and probes used in this study (Table 2), the detection work could not be carried out correctly, and the data were wrong.

Response 5: Thank you for pointing this out. We agree with this comment. An oversight occurred when writing this section and primers used in other procedures were included. This has now been corrected and the primers and probes used are shown. The output data from the thermocycler are attached as documentary support.

Comments 6: Line 203: It needs to check the description is accurate: p=<0.05.

Response 6: Thank you for pointing this out. We agree with this comment; the statistical connotation was not correct. Therefore, we have removed that connotation and expanded the information presented in lines 208-210. The changes made are highlighted in red in the revised manuscript.

Comments 7: The data in Table 5 requires proper analysis in manuscript.

Response 7: Thank you for pointing this out. We agree with this comment. Consequently, we have expanded the analysis of Table 5 in the manuscript's discussion, considering the elements presented in the table in greater depth.

Comments 8: Please review all statistical reporting in your manuscript. It is important that the statistical analyses used are suitable for your study and that all statistical reporting is correct.

Response 8: Thank you for pointing this out. We agree on the importance of verifying the presented methods and statistical data. Therefore, we have conducted a careful review of the statistics performed and presented. The adjustments made are highlighted in red in the resubmitted manuscript.

4. Response to Comments on the Quality of English Language

Point 1: The manuscript should be reviewed by a native English speaker who is familiar with scientific writing.

Response 1: Thank you for your suggestion. We would like to inform you that the manuscript has already been reviewed by an expert in scientific writing in English. We appreciate your feedback and are open to any specific recommendations for further improvement.

Reviewer 2 Report

Comments and Suggestions for Authors

Porcine circovirus type 2 infection is an important pathogen that endangers the normal development of the global swine industry, due to the different environment and breeding conditions in different regions, the prevention and control measures are also different, so it is of great significance to understand the PCV2 infection situation and epidemic risk factors in different regions for local disease prevention and control. This study evaluated pig production conditions in Jalisco, Mexico, and analyzed the factors associated with PCV2 infection by correlating with the epidemiological data obtained, providing a basis and technical guidance for the prevention and control of porcine circovirus type 2. However, there are still some issues in the research process of this paper. The major points are as follows:

1. There is a lack of introduction to the "production stage" clusters, and it is recommended to tabulate the four clusters.

2. Lack of data tables for analysis of frequency of PCV2-positive pools by production stage in Closed-cycle (CC) farms and multisite (MS) farms.

3. Revise the third sentence of the summary.

4.Confirm that the full text statement is grammatically correct, for example, line 33 “is it the smallest swine virus with a diameter ranging from 17 to 20 nanometers.

5.Ensure consistent line weights for Table 3 and Table 4.

6.The discussion in this article should be more in-depth and comprehensive. The current content is insufficient, so it is suggested to enhance it for better understanding and insight.

Comments on the Quality of English Language

The manuscript must be polished by a professional English language expert.

Author Response

Comments 1: There is a lack of introduction to the "production stage" clusters, and it is recommended to tabulate the four clusters.

Response 1: Thank you for pointing this out. We agree with this comment. Therefore, we have reorganized the information in the Materials and Methods section to clarify the age range with the analyzed production stages. We have also standardized the tabulations in the tables within the document. The changes made are highlighted in red in the adjusted tables and in the text on lines 105-117 of the resubmitted manuscript.

Comments 2: Lack of data tables for analysis of frequency of PCV2-positive pools by production stage in Closed-cycle (CC) farms and multisite (MS) farms.

Response 2: We appreciate your observation. We fully agree with this comment, and as a result, we have included the requested information in Table 4. The changes made are highlighted in blue within the corresponding table in the resubmitted manuscript.

Comments 3: Revise the third sentence of the summary.

Response 3: Thank you for pointing this out. We agree with this comment. Therefore, we have improved the wording of the highlighted sentence as well as the preceding ones. The changes made are shown in red in the resubmitted manuscript.

Comments 4: Confirm that the full text statement is grammatically correct, for example, line 33 “is it the smallest swine virus with a diameter ranging from 17 to 20 nanometers.”

Response 4: Thank you for pointing this out. We agree with this comment. Therefore, we have conducted a thorough grammatical review and correction throughout the document. The changes made are highlighted in red in the resubmitted manuscript.

Comments 5: Ensure consistent line weights for Table 3 and Table 4.

Response 5: Thank you for pointing this out. We agree with this comment. Therefore, we have revised and standardized the border thickness across all tables.

Comments 6: The discussion in this article should be more in-depth and comprehensive. The current content is insufficient, so it is suggested to enhance it for better understanding and insight.

Response 6: Thank you for pointing this out. We agree with this comment. Therefore, we have expanded the discussion; the adjustments can be seen in red in the revised manuscript.

4. Response to Comments on the Quality of English Language

Point 1: The manuscript must be polished by a professional English language expert.

Response 1: Thank you for your suggestion. We would like to inform you that the manuscript has already been reviewed by an expert in scientific writing in English. We appreciate your feedback and are open to any specific recommendations for further improvement.

Round 2

Reviewer 2 Report

Comments and Suggestions for Authors

The authors have provided a positive response based on the concerns raised.

Comments on the Quality of English Language

The current language is appropriate.